# ALIGNING REASONING LLMS FOR MATERIALS DISCOVERY WITH PHYSICS-AWARE REJECTION SAMPLING

## ABSTRACT

AI-driven materials discovery that couples automated experimentation with algorithmic decision-making requires process aware recipe to property predictors that are accurate, calibrated, and physically admissible. We approach this as a reasoning problem with large reasoning models (LRMs). To instill reasoning capability into language models, we curate reasoning traces from a teacher model to train a student model. However, most training pipelines select reasoning traces using binary correctness or learned preference signals that poorly reflect physical admissibility. We introduce *Physics-aware Rejection Sampling (PaRS)*, a training-time trace selection scheme that favors traces consistent with fundamental physics and numerically close to targets, with lightweight halting to control compute. We instantiate our framework with a large student model fine-tuned on traces synthesized by a larger teacher model, and evaluate under matched token budgets against various rejection sampling baselines. Our method improves accuracy and calibration, reduces physics-violation rates, and lowers sampling cost relative to baselines. These results indicate that modest, domain-aware constraints combined with trace-level selection provide a practical path toward reliable, efficient LRMs for process-aware property prediction and closed-loop materials design.

## 1 INTRODUCTION

A central goal in materials discovery is to compress the experimental loop by coupling automated experimentation with algorithmic decision making. Within this loop, property prediction is the core module. Accurate models that map composition, structure, and process/recipe variables to target properties (e.g., materials properties or device-level figures of merit) convert combinatorial exploration into tractable optimization and enable closed-loop design via bayesian optimization, provided that the models expose calibrated uncertainty and lightweight physics constraints to keep proposals physically admissible (Stach et al., 2022; Flores-Leonar et al., 2024; Frazier, 2018; Sabanzagil et al., 2025; Jacobs et al., 2024; Varivoda et al., 2022). Prior work has largely targeted properties from composition or crystal structure (Xie & Grossman, 2018; Chen et al., 2019) and, more recently, has explored text- or instruction-conditioned surrogates with LLMs (Ndayishimiye et al., 2025); yet scaling these predictors to process-aware, recipe to property tasks at the device level remains challenging (Liu et al., 2022; Lu et al., 2023; Xie et al., 2024).

Recently, large reasoning models (LRMs)—language models trained and/or reinforced to produce reliable reasoning traces—have shown dominant performance in diverse areas such as math, coding and scientific QA (Guo et al., 2025; Jaech et al., 2024; Yang et al., 2025). Their step-wise reasoning capabilities are a natural fit for recipe to property prediction, where multi-step physical arguments along the classic *chemical composition → process → micro-structure → property* chain are essential and well established in integrated computational materials engineering (National Research Council, 2008). Despite the promise, the leveraging LRMs on property prediction task remains underexplored relative to LLM and naive knowledge extraction (Zhao et al., 2024; Ndayishimiye et al., 2025). In this paper, we study how to train LRMs that reason effectively about materials recipes and output numerically correct, physically grounded properties.

A prevailing strategy for training LRMs to reason is to use filtered or re-weighted training signals based on the quality of teacher generated traces, complemented by test-time scaling (Muennighoff et al., 2025). Concretely, models are fine-tuned on self-generated rationales kept only when they

reach correct outcomes (Zelikman et al., 2022b; Yuan et al., 2023), or on samples ranked by a learned reward/verifier (Dong et al., 2023; Cobbe et al., 2021; Zheng et al., 2023b), or they aggregate multiple samples at decoding (Wang et al., 2022). These methods work as standard building blocks for LRMs along with SFT-only pipelines and RL-style post-training (Xu et al., 2025; Chen et al., 2025).

We argue that training property prediction LRMs from generated reasoning traces requires more sophisticated, physically grounded rejection sampling. Two characteristics of this task drive the need: *(1) High combinatorial design space.* The composition, process, structure, property chain creates high-dimensional, multi-mechanism spaces; inverse maps are often non-unique, yielding traces that seem plausible yet are scientifically incorrect (Xiang et al., 1995; Takeuchi et al., 2002; Ren et al., 2021; Liu et al., 2018; Yang et al., 2022). Therefore, effective learning requires sufficient exploration that searches both the design and reasoning trace space. *(2) Physically grounded outputs.* Targets are physical quantities whose magnitudes are constrained by physics and even small numeric deviations matter. Filters must therefore enforce admissible ranges and physical constraints from conservation laws and constitutive relations rather than rely solely on binary correctness.

Motivated by these challenges, we propose *Physics-aware Rejection Sampling (PaRS)*, a domain-tailored approach to optimize reasoning traces. Unlike prior methods that depend on binary correct-ness or learned reward models, our method couples rejection sampling with task-native, continuous error metrics derived from wet-lab experiments. Concretely, for each device recipe, we sequentially generate candidate traces, accepting the first trace that satisfies physics-aware acceptance gates and halting sampling early when further candidates show negligible variance or improvement.

We adopt Qwen3-32B [1] as the backbone model, fine-tuned via supervised fine-tuning (SFT) on internal prompts, with teacher reasoning traces synthesized from Qwen3-235B [2] (Yang et al., 2025). We benchmark against various rejection sampling methods under matched token budgets. Empirically, our method achieves the highest overall accuracy and calibration, while also delivering superior compute efficiency compared to existing baselines.

Our contributions are threefold.

- We formulate recipe to property prediction as a reasoning task with LRMs where physics-aware verification is essential.
- We propose novel physically grounded rejection sampling for optimizing reasoning traces, introducing the combination of powerful gating and halting techniques.
- We conduct (1) a teacher-side ablation, comparing our physics-aware sampler against six baselines in terms of trace accuracy and sampling efficiency, and (2) a student-side evaluation, fine-tuning an open-source LRM to demonstrate consistent gains in accuracy, calibration, and compute efficiency over all baselines.

## 2 RELATED WORK

**LLMs for materials design**    Recent advances in large language models (LLMs) have demonstrated strong generalization capabilities in materials design, drawing on interdisciplinary knowledge from chemistry, physics, and engineering (Miret & Krishnan; Jia et al., 2024). Beyond general-purpose LLMs, domain-specific models such as MatSciBERT (Gupta et al., 2022), MatBERT (Wan et al., 2024), and MELT (Kim et al., 2024) have been trained on large-scale materials science corpora, successfully capturing fundamental concepts that link structure, properties, processes, and perfor-mance. A key step in the materials design pipeline is accurate property prediction from symbolic representations, which serves as a surrogate for expensive experiments and enables rapid candidate screening. Leveraging their ability to process unstructured scientific data, recent studies have applied LLMs to this task without the need for elaborate feature engineering. For example, LLM-Prop (Niy-ongabo Rubungo et al., 2025) employs LLMs to predict crystalline material properties, while Li et al. (Li et al., 2025) integrate LLMs with graph neural networks for improved prediction accuracy. In the context of quantum-dot materials, Choi et al. (Choi et al., 2025) developed LLM-based synthesis protocol generation and property prediction models, fine-tuned on proprietary synthesis datasets.

---

[1] https://huggingface.co/Qwen/Qwen3-32B

[2] https://huggingface.co/Qwen/Qwen3-235B-A22B

LLMs have also been explored as surrogate models in optimization frameworks. LLAMBO (Liu et al.) utilizes the exploration capability of LLMs within Bayesian optimization, and BOPRO (Agarwal et al., 2025) incorporates LLM-based search strategies that exploit evolving uncertainty estimates to propose promising candidates in each iteration, thereby accelerating the discovery of globally optimal solutions.

**Large Reasoning Models (LRMs)**  The paradigm of next-token prediction has undergone a significant shift with the introduction of "thought" concept—a sequence of intermediate steps representing a model's internal reasoning process (Jaech et al., 2024; Guo et al., 2025; Yang et al., 2025; Muennighoff et al., 2025). This innovative approach enables LLMs to mimic complex human reasoning, such as reflective thinking and tree search. Chain-of-Thought (CoT) prompting (Wei et al., 2022) initially demonstrated that few-shot rationales could unlock sophisticated reasoning in LLMs, while Self-Consistency (Wang et al.) further enhanced reliability by marginalizing over multiple reasoning paths. Tree-of-Thoughts (ToT) (Yao et al., 2023) later reframed inference as a search over partial thought sequences, incorporating look-ahead and backtracking. Beyond prompting, process supervision has emerged as a powerful technique, training step-level verifiers or reward models to guide the reasoning process. This approach has been shown to outperform models trained with outcome-only labels, particularly in mathematical reasoning tasks (Lightman et al., 2023). More recently, GPT-o1 (Jaech et al., 2024), Qwen3 (Yang et al., 2025), and DeepSeek-R1 (Guo et al., 2025) have popularized modern Large Reasoning Models (LRMs) by integrating long-form thinking with process supervision, RL-based post-training, and test-time scaling. Building on these foundational advances, our work adapts these reasoning mechanisms to the domain of materials design, specifically targeting physically grounded recipe to property prediction.

**Rejection sampling for LLMs**  Rejection sampling is widely recognized as an effective data-filtering method that promotes higher-quality supervision. In the context of RLHF and preference-optimization, it narrows multiple generated outputs per prompt to only the high quality responses, as determined by a reward model during post-training adjustments. For example, RAFT (Dong et al.) aligns generative models efficiently by using a reward model and abundant candidate samples; it discards outputs demonstrating undesired behaviors and fine-tunes the model solely on the selected high-quality subset. Building on this, Reinforce-Rej (Xiong et al., 2025) proposes a minimalist policy-gradient extension that filters out both entirely incorrect and entirely correct samples, enhancing stability and efficiency. In reasoning tasks, STAR-like models (Zelikman et al., 2022a; Hosseini et al., 2024; Koh et al., 2025) eliminate expensive human annotations by employing a self-taught reasoning loop—generating chain-of-thought traces, self-verifying correctness, and fine-tuning only on reliable examples. Additionally, Rejection Sampling Fine-Tuning (RFT) (Yuan et al., 2024) enhances mathematical reasoning by incorporating model-generated reasoning traces filtered for correctness into training.

## 3 METHOD

### 3.1 TASK DEFINITION

Quantum dot light-emitting diodes (QD-LEDs) are electroluminescent devices that use colloidal semiconductor quantum dots as the emissive layer, offering narrowband spectra and composition tunable color (Shirasaki et al., 2013a; Li et al., 2024). In practice, closed-loop materials design must optimize figures of merit measured at the device level—e.g., peak external quantum efficiency and operational stability—because these metrics ultimately determine application viability for emissive displays and lighting (e.g., televisions, monitors) (Bang et al., 2021; Huang et al., 2020; Li et al., 2024). We therefore frame inputs as complete device recipes: a multi-layer stack typically consist of anode/ITO, hole-injection layer (HIL), hole-transport layer (HTL), quantum-dot emitting layer (EML), electron-transport layer (ETL), electron-injection layer/cathode with per layer materials and process parameters. Each layer records identifiers (material, formulation), geometric parameters (e.g., thickness), and process variables (e.g., solution concentration, spin profile, bake/anneal temperature and duration, atmosphere), along with post-process steps (e.g., UV–ozone, plasma, solvent rinse). See Figure 1 for a recipe example.

We formulate QD-LED device property prediction as a reasoning LLM task. Let $\mathcal{D} = \{(x_i, y_i)\}_{i=1}^{N}$ where $x_i$ is a QD-LED recipe as above and $y_i \in \mathbb{R}$ is a device-level target; in this work we focus

```
QD-LED recipe:
  substrate:
    type: ITO/glass, thickness_nm: 150, rsheet_ohm_sq: 15,
    roughness_Rq_nm: 1.5
    pretreat: UV-ozone, 10 min; solvent rinse (IPA); 120 C annealing 10
    min
  stack:
    [HIL layer]
      substances: PEDOT:PSS (AI4083), thickness_nm: 40, filtration_um:
    0.45
      work_function_eV: 5.1, process: spin (4000 rpm, 60 s -> 8000 rpm, 5
    s ramp)
      annealing (150 C, 10 min, air)
    [HTL layer]
      substances: Poly-TPD, thickness_nm: 20, HOMO_eV: -5.2
      solution: 8 mg/mL, chlorobenzene (99.8%, anhydrous), filtration_um:
    0.2
      process: spin(3000 rpm, 45 s); annealing(120 C, 10 min, N2)
    [EML layer]
      substances: CdSe/ZnS core/shell QDs (green), emission_peak_nm: 525,
      FWHM_nm: 22, core_diameter_nm: 5.5, ligand_primary: oleic acid /
    oleylamine,
      solution_conc_mg_mL: 25, solvent: octane (anhydrous, <10 ppm H2O)
      filtration_um: 0.2 PTFE, target_areal_density_ug_cm2: 40,
    thickness_nm: 25,
      process: spin (2000 rpm, 30 s); annealing (80 C, 5 min),
    film_roughness: 1.8
      PLQY_solution_fraction: 0.92, PLQY_film_fraction: 0.80
    [ETL layer]
      substances: ZnO nanoparticles (sol-gel/colloidal),
    mean_particle_diam_nm: 5
      thickness_nm: 30, solution: 10 mg/mL, isopropanol
      process: spin(3000 rpm, 30 s); annealing (90 C, 5 min, N2)
        ...
```

Figure 1: Structured QD-LED recipe example.

```
You are a world-class expert in quantum-dot light-emitting-diode (QD-LED)
    device
physics and fabrication.

<Query QD-LED recipe>

TASK: Predict external quantum efficiency for a QD-LED device fabricated
    by the
query recipe.

Final output format (only json output)
Please provide your final report in a structured JSON format.
{
  "answer": <PREDICTED_VALUE> %
}
```

Figure 2: `Prompt` for the property prediction task with large reasoning models (LRMs).

on max external quantum efficiency (i.e., $y_i = \text{EQE}_{\max}$). Given a task-specific `prompt` (See Figure 2) and recipe $x_i$, a large reasoning model $f_\theta$ produces a reasoning trace $\tau$ along with a numeric prediction $\hat{y}$:

$$f_\theta(\texttt{prompt}, x_i) \rightarrow (\tau, \hat{y}). \tag{1}$$

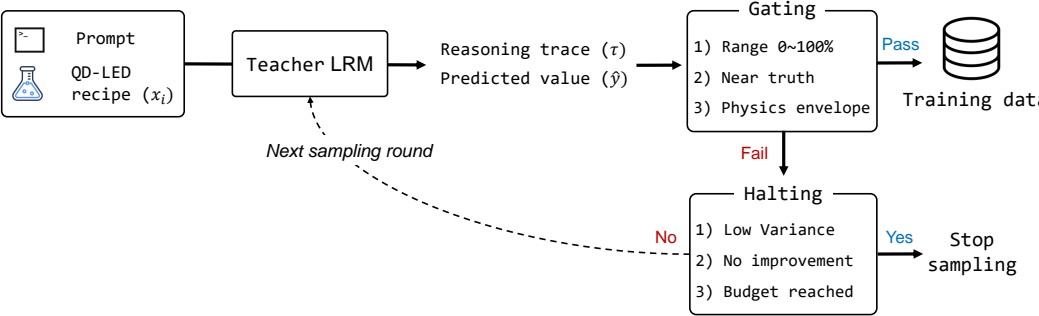

Figure 3: `PaRS` workflow: the teacher generates a mini-batch of candidates; a candidate is accepted only if it passes gates (range, near-truth tolerance, physics envelope). If none pass, halting checks decide whether to stop or raise temperature and continue to next sampling round. Accepted traces supervise the student model as training data.

## 3.2 PaRS: Physics-aware Rejection Sampling

For training LRMs, the supervision signal extends beyond the final answer to include the sampled reasoning traces themselves. Achieving high-quality supervision for reasoning LLMs therefore requires rejection sampling to filter out suboptimal traces. As illustrated in Figure 3, we propose *Physics-aware Rejection Sampling (PaRS)*, which integrates physics-aware gating and halting mechanisms to optimize reasoning traces

**Physics-aware Gating** For each recipe $x_i$, we generate reasoning traces sequentially from the teacher model up to $K$. We accept the first sample that satisfies all of following acceptance gates:

$$\hat{y}^{(k)} \in [0, 100], \tag{2}$$

$$\left|\hat{y}^{(k)} - y_i\right| \leq \varepsilon_{\text{MAE}}, \tag{3}$$

As $\text{EQE}_{\max}$ is reported as a percentage, Eq. (2) enforces range consistency. Unlike categorical correctness filters (Zelikman et al., 2022b; Yuan et al., 2023; Dong et al., 2023; Wang et al., 2022; Muennighoff et al., 2025), Eq. (3) uses a continuous error against wet-lab ground truth, yielding richer learning signals.

With external quantum efficiency (EQE) as the prediction target, we can define an empirical upper bound $U(x)$ as follows. EQE is commonly factorized as $\text{EQE} = \eta_{\text{out}} \cdot \eta_{\text{rad}} \cdot \gamma$ with $\eta_{\text{out}} \leq 1$ and $\gamma \leq 1$ by definition. In regimes where solid-state PLQY limits the radiative yield, the conservative relation $\eta_{\text{rad}} \leq \text{PLQY}$ holds, implying $\text{EQE} \leq \text{PLQY}$ (Shirasaki et al., 2013b; Gather & Reineke, 2015; OE2, 2014). PLQY (photoluminescence quantum yield) denotes the ratio of emitted to absorbed photons; we use the solid-state PLQY of the EML under device-relevant conditions. We therefore define an empirical, recipe-specific upper bound as $U(x) = U_{\text{PL}}(x)$, where $U_{\text{PL}}(x)$ is instantiated by the highest measured film photoluminescence quantum yield of the emissive layer in the recipe $x$. We add this upper bound to the acceptance gates:

$$\hat{y}^{(k)} \leq U(x_i) \tag{4}$$

It prevents reasoning traces that predict physically implausible overshoots for target property. If no sampled candidate satisfies Eq. (2)–(4), we discard the example.

**Adaptive Halting** We sample in mini-batches of size $b$ and proceed round by round until the total budget $K_{\max}$ is exhausted. In round $r$ ($r = 1, 2, \ldots$), we draw exactly $b$ candidates $\{(\tau^{(r,j)}, \hat{y}^{(r,j)})\}_{j=1}^{b}$ at temperature $T_r$. For each candidate, we apply the acceptance gates in Eqs. (2)–(4). If any candidate passes, we accept the earliest passing one and terminate sampling.

If no candidate in the mini-batch passes, we apply two halting checks before proceeding. (i) *Variance-based halting* (from round 1): stop when the within-batch error variance falls below a threshold, indicating insufficient diversity to justify further exploration. (ii) *Improvement-based halting* (from round 2): stop when the best error in the current round fails to improve over the previous round by

Table 1: Analysis of reasoning trace selection. All traces are generated with QWEN3-235B. Higher LLM-as-a-Judge score and lower MAE is better. $K_{avg}$ is the average number of generated reasoning traces per prompt. Our Halting logic yields $K_{avg}$=6.4 on average (fewer required tokens for selected trace). 'Budget→selected' counts teacher generates per prompt and the number retained after selection.

| Method | $K_{avg}$ | Budget (→ selected) | LLM-as-a-Judge (score, 0–10) | MAE |
|---|---|---|---|---|
| No sampling | 1 | $1 \rightarrow 1$ | 5.97 | 2.440 |
| Random sampling | 12 | $12 \rightarrow 1$ | 5.86 | 2.327 |
| Longest trace | 12 | $12 \rightarrow 1$ | 6.10 | 2.274 |
| Self-consistency | 12 | $12 \rightarrow 1$ | 5.78 | 1.829 |
| LLM-as-a-Judge | 12 | $12 \rightarrow 1$ | 6.55 | 2.223 |
| Multi-sampling | 12 | $12 \rightarrow 12$ | 5.89 | 2.356 |
| **PaRS (Ours)** | **6.4** | $12 \rightarrow 0.8^{\dagger}$ | **7.51** | **0.829** |

$^{\dagger}$ We drop the around 20% of sample that not passed our acceptance logic in Sec. 3.2 thus 0.8 traces kept per prompt on average.

at least a small margin. We also halt once the cumulative number of sampled candidates reaches $K_{max}$. If none of these conditions trigger, we increase $T_r$ and continue to the next round to encourage exploration. Refer Appendix A.1 for the details of halting methods.

## 4 EXPERIMENTS

### 4.1 BASELINES

We curate 11k QD-LED device dataset and split into 10k for training and 1k for testing. We construct prompts from all 11k dataset for a property prediction task and query 10k train prompts with QWEN3-235B to sample teacher reasoning traces. To ensure fair comparison, each reasoning trace has a same sampling budget of $K$=12, except for no sampling. We compare the following methods for selecting reasoning traces including our method.

1. **No sampling:** use the first generated trace.

2. **Random sampling:** uniformly sample one of the $K$ traces.

3. **Self-consistency aggregation:** select the trace whose final answer is closest to the median across all $K$ answers (Wang et al., 2022).

4. **Longest trace:** select the trace with the largest token length.

5. **LLM-as-a-judge:** score all traces with a larger judge model (DeepSeek-R1 [3]) and select the top-ranked trace (Zheng et al., 2023a). Refer Appendix. A.2 for details of the LLM-as-a-Judge prompting.

6. **Multi-sampling:** retain *all* $K$ traces as supervision (Guha et al., 2025).

7. **PaRS (ours):** mini-batch size $b$=4 with a temperature schedule $T \in \{0.6, 0.8, 1.0\}$ increasing by 0.2 per round. We set $\varepsilon_{MAE}$=1, $\varepsilon_{var}$=1 and $\delta_{imp}$=1 by analyzing the data distribution. If no candidate passes the gates within the budget, the example is discarded.

After rejection sampling, we fine-tune QWEN3-32B as the student model for a single epoch on the traces selected by each method, using AdamW (learning rate $2\times10^{-5}$) on 32×A100 (80 GB) GPUs; unless stated otherwise, all other training hyperparameters are shared across methods.

**Evaluation metrics** We report two groups of metrics. (1) Teacher-side trace quality: mean absolute error of the selected trace' prediction, and the average number of sampled traces per prompt to quantify the cost of constructing the selected traces. To further assess trace quality, we also employ a

---
[3]https://huggingface.co/deepseek-ai/DeepSeek-R1-0528

Table 2: QD-LED property prediction with QWEN3-32B trained on teacher traces selected by each method from QWEN3-235B. Lower MAE is better; higher $R^2$ and Spearman $\rho$ are better. Viol.% is computed on test predictions without post hoc clipping to $U(x)$. #train is the number of supervision traces used for SFT: multi-sampling retains all $K{=}12$ traces per example ($12\times$ number of train prompts), whereas our method keeps only candidates that pass our gates, resulting in fewer traces.

| Training data (prompt + reasoning trace) | # train | MAE | $R^2$ | Spearman $\rho$ | Viol. (%) |
|---|---|---|---|---|---|
| No sampling | 10 000 | 2.001 | 0.376 | 0.607 | 35.8 |
| Random sampling | 10 000 | 1.961 | 0.358 | 0.621 | 35.2 |
| Longest trace | 10 000 | 1.942 | 0.375 | 0.614 | 35.3 |
| Self-consistency | 10 000 | 1.933 | 0.377 | 0.629 | 36.8 |
| LLM-as-a-Judge | 10 000 | 1.889 | 0.408 | 0.667 | 35.4 |
| Multi-sampling | 120 000 | 1.984 | 0.335 | 0.632 | 36.6 |
| **PaRS (ours)** | 7980 | **1.808** | **0.424** | **0.705** | **27.7** |

larger LLM (DeepSeek-R1) as an external judge, providing external evaluation of reasoning quality on a 0–10 scale. (2) Student-side performance: MAE, $R^2$, Spearman's $\rho$, and a physics violation rate—the fraction of predictions that fall outside $[0, 100]$ or exceed the empirical upper bound $U(x)$ on the hold out test set.

For each test prompt, we run five independent inferences with the trained student models and take the median of the five predictions before computing MAE, $R^2$, and Spearman's $\rho$. For the physics violation rate, we evaluate the constraint indicator on each of the five predictions and report the average violation frequency across all ensemble. Details of the LLM-as-a-Judge prompting procedure are provided in Appendix A.2.

## 4.2 RESULTS

Our experiments show that PaRS effectively optimize the teacher' reasoning traces to induce reasoning capability for property prediction in student model. We present our findings in two parts: first, an analysis of the trace selection process itself and second, an evaluation of the trained student LRMs on the property prediction task.

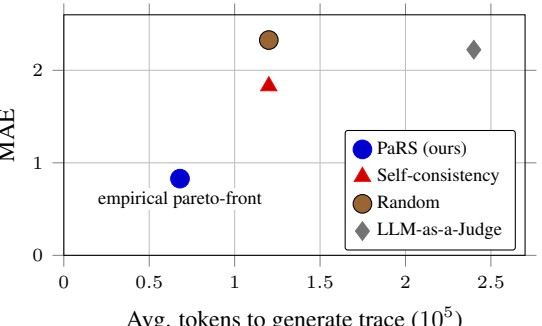

**Analysis of reasoning trace sampling** We evaluate rejection sampling strategies on the traces generated by the teacher QWEN3-235B. As summarized in Table 1, PaRS achieves lower prediction error while requiring fewer generations on average, placing it on the empirical quality–efficiency Pareto front (Fig. 4). A simple MAE gate could make low error appear trivial, yet an external LLM-as-a-judge that is not optimized for our metric still assigns PaRS the highest overall score. Self-

Figure 4: Compute–accuracy frontier for rejection sampling methods. Our approach achieves the lowest teacher MAE with substantially fewer required tokens, forming the empirical Pareto front. The x-axis shows average required tokens for generating reasoning trace per prompt and the y-axis shows teacher MAE. See Appendix A.3 for details.

consistency, which got the lowest MAE except for our method, receives the weakest judge score among the baselines. This gap suggests that generic preference signals do not fully align with physics-grounded proximity to ground truth. By combining physics-aware gates with temperature scheduling and early stopping, PaRS concentrates supervision on high-fidelity, physically admissible traces rather than on merely "good-looking" rationales.

Table 3: **Comparison of reasoning traces generate by three teacher models**. We calculate MAE for all traces by using all generated traces (seven traces per prompt in average). MAE from selected traces is computed only over selected traces by `PaRS`. Selected ratio is the fraction of generated traces accepted by textttPaRS.

| Teacher model | MAE for all traces | MAE for selected traces | Selected ratio |
|---|---|---|---|
| DeepSeek-R1-671B | 2.144 | 0.785 | 0.789 |
| Qwen3-235B | 2.327 | 0.829 | 0.798 |
| Qwen3-235B-FP8 | 9.078 | 0.812 | 0.811 |

Table 4: **Distillation results.** Each row fine tunes the same student (QWEN3-32B) on selected traces generated by the indicated teacher models with `PaRS`.

| Distiled models | # train | MAE | $R^2$ | Spearman $\rho$ | Viol. (%) |
|---|---|---|---|---|---|
| DeepSeek-R1-Distill-Qwen-32B | 7897 | 1.755 | 0.445 | 0.717 | 25.1 |
| Qwen3-235B-Distill-Qwen-32B | 7980 | 1.808 | 0.424 | 0.705 | 27.7 |
| Qwen3-235B-FP8-Distill-Qwen-32B | 8112 | 1.801 | 0.413 | 0.712 | 28.9 |

**Evaluation of distilled LRMs**  Training the student QWEN3-32B on traces selected by our method yields consistent gains in accuracy, correlation with ground truth, and physical admissibility, while using substantially fewer supervision traces than competing approaches (See Table 2). The LLM-as-a-Judge baseline is competitive in error and correlation but does not match the reduction in violations or the calibration gains achieved by `PaRS`. Retaining all sampled traces enlarges supervision volume yet underperforms, consistent with amplified label and reasoning noise when traces remain unfiltered. Recent work (Guha et al., 2025) finds that multi-sampling can help by preserving reasoning diversity, but in our task the same unfiltered diversity amplifies trace noise, increases physics violation rate and weakens calibration. `PaRS` reconciles these views by keeping diversity where it matters, namely multiple near correct physical pathways, while trimming supervision to numerically consistent traces that respect simple physics. The result is a better accuracy and admissibility with about $15\times$ fewer traces than the multi sampling baseline.

These improvements arise from aligning the acceptance rubric with a continuous, physically grounded target. The range check in Eq. (2) and the empirical envelope in Eq. (4) for target EQE suppress implausible overshoots. The continuous gate in Eq. (3) rewards numerical proximity to wet-lab measurements rather than binary correctness. Since the mapping from recipe to property is many to one and complex, enforcing zero error steers supervision toward outliers and collapses trace diversity. The tolerance $\varepsilon_{\mathrm{MAE}}$ instead admits near-correct and physically plausible traces, which preserves multiple valid pathways, improves calibration, and lowers violation rates. The halting logic further reduces redundant sampling and concentrates supervision on diverse, high-fidelity trajectories.

## 4.3 ANALYSIS

**Robustness of PaRS across teacher models**  To validate robustness of our method across the diverse teacher models, we sample reasoning traces with DeepSeek-R1-671B and Qwen3-235B-FP8 in addition to Qwen3-235B. MAE for all traces in Table 3 shows that smaller or low-precision teacher models produce noisier trace distributions than large models. However, after applying PaRS, the selected traces from small quantized teachers converge to the quality of those of large teachers: error and stability tighten, and the retained supervision is physically admissible and consistently high quality. In effect, `PaRS` equalizes teacher induced variability by filtering out unstable generations and preserving only high fidelity reasoning trajectories.

This equalization propagates to the student. Fine-tuning the same student on selected traces from different teachers yields similar MAE, correlation, and violation rates. The correctness ratio of selected traces also varies little across teachers, indicating that `PaRS` is broadly compatible with heterogeneous teacher models and precisions. Consequently, even smaller or low-precision teachers can provide supervision competitive with that from larger models.

**How trace correctness ratio shapes distillation performance** We study how the fraction of accepted traces (the "correctness ratio" under the tolerance gate $\varepsilon_{\mathrm{MAE}}$) relates to student performance (Fig. 5). Holding the dataset size and token budget fixed, increasing the acceptance rate $r$ monotonically lowers MAE and reduces physics violations. Interestingly, recent work (Muennighoff et al., 2025) shows that curated rationales and test-time scaling can yield strong results even when only 53% of traces are correct. We hypothesize that tasks such as math, coding, and QA exhibit binary correctness with explicit derivations, so trace correctness closely mirrors supervision quality. However, for property prediction, categorical correctness is a weak proxy for supervision quality because two traces may follow different discrete steps yet end within 1% of the true EQE, which is the signal the model must learn.

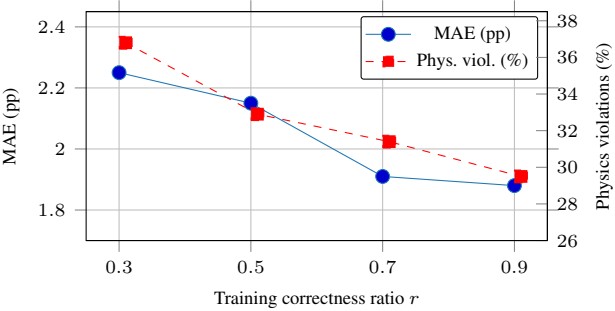

Figure 5: Effect of training correctness ratio on student performance. Training dataset size and inference token budget are fixed. Points show means over a 5-model ensemble on test prompts.

**Ablation on adaptive halting** We ablate the adaptive-halting mechanism in `PaRS` to assess the benefit of early stopping. As summarized in Table 5, enabling adaptive halting reduces the average number of sampled candidates from $K_{\mathrm{avg}}{=}8.9$ to $6.4$ and shortens compute time from 20.5,h to 15.1,h, with only a marginal change in accuracy. These results indicate that the variance- and improvement-based early stopping criteria are effective, low-overhead heuristics for pruning unproductive samples without materially degrading the quality of the retained traces. As the teacher model's parameters are fixed, persistently low variance and minimal improvement suggest limited capacity to produce faithful reasoning traces that recover ground-truth properties with physically plausible values.

Table 5: Ablation on adaptive halting. We use 10 nodes of $8 \times$ H100 (80G) to sample reasoning traces from the 10k prompts with Qwen3-235B and measure the compute time.

| Method | $K_{\mathrm{avg}}$ | MAE | Compute time (h) |
|---|---|---|---|
| PaRS w/o Adaptive Halting | 8.9 | 0.817 | 20.5 |
| PaRS w/ Adaptive Halting | 6.4 | 0.829 | 15.1 |

## 5 CONCLUSION

We cast recipe to property prediction for materials discovery as a reasoning problem and introduce *Physics-aware Rejection Sampling (PaRS)* to curate supervision signals that are numerically accurate, calibrated, and physically admissible. `PaRS` replaces binary correctness and generic reward models with domain-grounded gates—range checks, a recipe-specific physical envelope, and a continuous error tolerance and adds variance and improvement-based halting. Instantiated with a QWEN3-235B teacher and a QWEN3-32B student on QD-LED device recipes, `PaRS` consistently optimize teacher-trace quality and yields student LRMs with lower MAE, higher correlation, better calibration, and markedly fewer physics violations. These gains are achieved with substantially less sampling, indicating a favorable quality–efficiency trade-off. While our experiments focus on EQE in QD-LEDs, the framework naturally extends to other device-level properties (e.g., lifetime, luminance) and to materials systems beyond optoelectronics. An important next step is to evaluate `PaRS` at larger scale and across diverse domains. We also envision coupling `PaRS` with RL-based adaptive exploration to enable closed-loop recipe design, where models not only predict reliably but also guide autonomous exploration of materials space under explicit physical guarantees.

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

# A  IMPLEMENTATION DETAILS

## A.1  SAMPLING SCHEDULE

For an example $(x_i, y_i)$ and rounds $r = 1, 2, \ldots$ with mini-batch indices $j = 1, \ldots, b$, draw candidates $(\tau^{(r,j)}, \hat{y}^{(r,j)})$ and define the per-candidate error $e_{r,j} := |\hat{y}^{(r,j)} - y_i|$. For each round, let $\bar{e}_r := b^{-1} \sum_{j=1}^{b} e_{r,j}$, $s_r^2 := (b-1)^{-1} \sum_{j=1}^{b} (e_{r,j} - \bar{e}_r)^2$, and $e_r^\star := \min_{1 \le j \le b} e_{r,j}$; for $r \ge 2$, define the improvement $\Delta_r := e_{r-1}^\star - e_r^\star$. The total candidate budget is $K_{\max}$, and $T_r$ denotes the sampling temperature in round $r$.

A candidate is accepted if it satisfies all gates in Eqs. (2)–(4), namely $\hat{y}^{(r,j)} \in [0, 100]$, $|\hat{y}^{(r,j)} - y_i| \le \varepsilon_{\text{MAE}}$, and $\hat{y}^{(r,j)} \le U(x_i)$. If multiple candidates pass within the same round, accept the one with the smallest $j$ and terminate for that example.

If no candidate is accepted in round $r$, the procedure halts early under any of the following conditions: variance small enough, $s_r^2 \le \varepsilon_{\text{var}}$ (available from $r = 1$); lack of improvement, $\Delta_r \le \delta_{\text{imp}}$ (available from $r = 2$); or budget exhausted, $r\, b \ge K_{\max}$.

The temperature follows a capped, monotone increase to encourage exploration, for example $T_{r+1} = \min\{T_{\max}, \gamma T_r\}$ with $\gamma > 1$, or $T_{r+1} = \min\{T_{\max}, T_r + \Delta T\}$ with $\Delta T > 0$, starting from $T_1 = T_{\min}$.

The overall procedure is: at round $r = 1$, sample $b$ candidates at temperature $T_r$ and apply the acceptance gates; if none pass, compute $s_r^2$, $e_r^\star$, and (for $r \ge 2$) $\Delta_r$; if no halting condition triggers, update the temperature according to the schedule and continue to $r+1$. Discard the example if no candidate is accepted before the budget is spent.

## A.2  PROMPT FOR LLM-AS-A-JUDGE

We use DEEPSEEK-R1-0528-671B as the LLM-as-a-Judge. The prompt in Fig. 6 instructs the judge to evaluate synthesized reasoning traces against five rubrics and to return a numeric score for each rubric on a 0–10 scale with a brief justification. For the *LLM-as-a-Judge* selection baseline, we score all candidate traces generated for a prompt, compute a composite judge score by averaging the five rubric scores, and select the best one per prompt. For the summary metric reported in Table 1, we evaluate the set of traces selected by each method with the same judge prompt. We compute the composite score for each trace as the mean over the five rubrics and then report the mean across all evaluated prompts. This yields a single 0–10 score per method that is comparable across rejection sampling methods.

## A.3  TOKEN ACCOUNTING FOR TRACE SELECTION

For each prompt, let $T_{\text{teach,in}}$ and $T_{\text{teach,out}}$ denote the teacher's input tokens and *average* output tokens per generated trace. Let $K$ be the generation budget, $G$ the random number of traces actually generated before acceptance or budget exhaustion, and $K_{\text{avg}} := \mathbb{E}[G]$. Let $T_{\text{select}}$ capture any extra token cost due to a selection pass (if present). Finally, let $r_{\text{acc}} \in [0, 1]$ be the probability that a prompt yields at least one accepted trace.

**Expected tokens per prompt**  The expected token cost per prompt, regardless of whether a trace is accepted, is

$$\mathbb{E}[\text{tokens per prompt}] = K_{\text{avg}}\big(T_{\text{teach,in}} + T_{\text{teach,out}}\big) + T_{\text{select}}. \tag{5}$$

For offline selection methods (random, self-consistency, and ours), the selection pass is negligible, so $T_{\text{select}} \approx 0$. By contrast, LLM-AS-A-JUDGE performs an additional inference pass over the concatenated set of generated traces. Approximating the judge pass as comparable in length to the teacher pass yields

$$\mathbb{E}[\text{tokens per prompt}]_{\text{judge}} \approx 2\, K_{\text{avg}}\big(T_{\text{teach,in}} + T_{\text{teach,out}}\big). \tag{6}$$

**Expected tokens per accepted trace**   When some prompts produce no accepted trace, it is useful to normalize by the acceptance rate $r_{\text{acc}}$. The expected tokens per accepted trace are

$$\mathbb{E}[\text{tokens per accepted}] = \frac{\mathbb{E}[\text{tokens per prompt}]}{\mathbb{E}[\text{accepted traces per prompt}]} \approx \frac{K_{\text{avg}}\big(T_{\text{teach,in}} + T_{\text{teach,out}}\big) + T_{\text{select}}}{r_{\text{acc}}},$$

(7)

with the judge variant obtained by substituting Eq. (6) into Eq. (7).

Under online acceptance (our method), generation halts immediately upon acceptance or when the budget $K$ is reached. Thus $G$ follows a truncated geometric-like process, and $K_{\text{avg}}$ reflects both early acceptance on easy prompts and full-budget usage on hard prompts. Methods that commit to a fixed $K$ without early stopping have $K_{\text{avg}} \approx K$.

In compute versus accuracy frontiers (e.g., Fig. 4), the x-axis reports the per-prompt token cost in Eq. (5). When comparing methods with materially different $r_{\text{acc}}$, we additionally report the per-accepted-trace cost using Eq. (7).

As a concrete example, suppose $T_{\text{teach,in}}{=}900$, $T_{\text{teach,out}}{=}2000$, $K_{\text{avg}}{=}6.4$, and $r_{\text{acc}}{=}0.8$. Then the per-prompt cost for offline selection is $6.4 \times (900 + 2000) = 18{,}560$ tokens. The judge variant is about $2 \times 18{,}560 = 37{,}120$ tokens per prompt. Normalizing by acceptance rate, the per-accepted-trace costs are approximately $23{,}200$ (offline) and $46{,}400$ (judge).

```
<Prompt>: Device recipe
<Response>: Model's reasoning trace + final prediction.

# Role
You evaluate QD-LED EQE prediction responses (especially reasoning trace)
    quality with following rubric. Judge only against the provided
    device prompt.

# Scoring rubric (0~10)
1. Groundedness to Prompt (0~2.5): Quote prompt substrings for all used
    parameters; mark extra info as Assumption.
- 0.0~0.5: Largely ungrounded; few/no quotes; multiple unstated details.
- 0.6~1.3: Some quotes, but several parameters not cited; occasional
    unstated claims
- 1.4~2.0: Mostly grounded; 1-2 minor misses; assumptions called out but
    one is vague
- 2.1~2.3: Fully grounded with trivial omissions only
- 2.4~2.5: Every device parameter quoted; zero unstated details

2. Causal Reasoning Quality (0~2.0): Link given factors -> mechanisms ->
    EQE impact; separate Given / Inference / Implication.
- 0.0~0.4: Descriptive or hand-wavy; leaps from factors to EQE without
    mechanism.
- 0.5~1.0: Some correct factor->effect links but gaps and mixing of Given
    /Inference.
- 1.1~1.5: Coherent chains for most factors; clear separation with one
    notable gap
- 1.6~1.8: Mechanism-first, no unjustified jumps; discusses main loss
    channels
- 1.9~2.0: Exemplary: prioritizes the limiting mechanism.

3. Numerical & Unit Discipline (0~2.0): Show steps; keep %/nm/eV
    consistent; sensible rounding of final EQE.
- 0.0~0.4: Arithmetic or unit errors ; missing key steps.
- 0.5~1.0: Mostly correct; one error or unit slip.
- 1.1~1.5: Correct math; consistent units; minor omission .
- 1.6~1.8: Fully worked steps (e.g., IQE x outcoupling); sanity checks.
- 1.9~2.0: Clean, reproducible pipeline; precision noted where relevant.

4. Assumption Quality (0~2.0): Assumptions explicit, minimal, non-
    contradictory, each briefly justified.
- 0.0~0.4: Many hidden or contradictory assumptions.
- 0.5~1.0: Several assumptions; some lack justification.
- 1.1~1.5: Only necessary assumptions; short, credible justifications.
- 1.6~1.8: Minimal & well-justified; references common baselines.
- 1.9~2.0: Parsimonious and transparent; each assumption tied to its EQE
    impact; brief sensitivity note if applicable.

5. Clarity & Structure (0~1.5): Use sections: Given / Assumptions /
    Reasoning / Result; keep high signal-to-noise.
* 0.0~0.3: Disorganized; sections missing; EQE result absent or hard to
    find.
* 0.4~0.7: Sections present but uneven; some redundancy; result line
    imprecise.
* 0.8~1.1: Clear sections; stepwise logic; minor verbosity or formatting
    slips.
* 1.2~1.3: Crisp, concise, well-formatted; Result line prominent.
* 1.4~1.5: Polished, minimal, easy to audit; bullets/tables used
    judiciously.
```

Figure 6: `Prompt` for evaluating reasoning traces with DeepSeek-R1. We report the sum of average score of the five metrics to Table 1.

