# OpenReview forum: "Aligning Reasoning LLMs for Materials Discovery with Physics-aware Rejection Sampling"
_ICLR.cc/2026/Conference — ICLR 2026 Conference Withdrawn Submission_

### Official Review · Reviewer_oa7n · 2025-10-25

**Soundness:** 2
**Presentation:** 2
**Contribution:** 2
**Rating:** 2
**Confidence:** 1

**Summary:**

The paper trains a small reasoning model for materials discovery by using a larger model to generate reasoning traces and then perform physics-aware filtering (PaRS) to select “valid” traces for distillation. Experiments on QD-LED device property prediction show improved MAE and lower violation rates relative to simple rejection sampling baselines.

**Strengths:**

The paper is clearly written and the problem of physics-admissible reasoning for materials tasks is practically relevant.

The evaluation protocol is relatively thorough within the chosen domain (teacher-side + student-side metrics).

Physics-based rejection constraints are reasonable and give some domain grounding compared with correctness-only filtering.

**Weaknesses:**

Core idea is conceptually straightforward and incremental
The pipeline is essentially:
(big LRM → generate → filter → distill to small LRM).
This paradigm already appears widely in reasoning LLM alignment (e.g., rejection sampling + SFT/RFT/RL), and the physics-aware gating here amounts to domain-specific heuristics rather than a new training principle.

Technical novelty is limited
The PaRS mechanism is a direct instantiation of rejection sampling with thresholding; there is no new modeling component, no new learning objective, and no new algorithmic insight beyond domain-informed gating.

**Questions:**

see weakness.

Due to my limited familiarity with materials discovery, I cannot fully assess the domain significance of the claimed contribution beyond the methodology itself.

---

### Official Review · Reviewer_9TLb · 2025-10-28

**Soundness:** 2
**Presentation:** 3
**Contribution:** 2
**Rating:** 2
**Confidence:** 3

**Summary:**

This paper tackles recipe-to-property prediction for materials discovery by training Large Reasoning Models on QD-LED external quantum efficiency data. The key observation is that existing rejection sampling methods (binary correctness, learned rewards) poorly suit regression tasks with physical constraints. The main contribution is Physics-aware Rejection Sampling (PaRS), which filters teacher-generated reasoning traces using three gates: range validity, continuous error tolerance , and a PLQY-derived physical upper bound. Adaptive halting reduces sampling cost via variance and improvement checks.

**Strengths:**

1. Originality:
 Adapting rejection sampling to continuous regression with physics constraints is reasonable. The continuous error tolerance + PLQY bounds combination is novel for LRM training. Adaptive halting based on variance/improvement is a practical addition.

2. Quality:
Solid experimental design—six baselines with matched token budgets. External LLM-as-judge adds independent evaluation. Ablations across three teacher models show robustness. Analysis of correctness ratio and halting provides practical insights.

3. Clarity:
Well-written with clear motivation. Problem setup is concrete (Figure 1 recipe example).

**Weaknesses:**

1. Limited methodological novelty: The workflow is standard teacher-student distillation with hand-crafted filtering rules. The only modification is replacing binary correctness with three domain-specific gates (range + tolerance + PLQY bound).

2. Missing hyperparameter analysis: Section 4.1 states "We set εMAE=1, εvar=1 and δimp=1 by analyzing the data distribution" but provides no sensitivity analysis. These three parameters directly control the gating and halting mechanisms—how robust are results to different choices?

3. Single-task evaluation: All experiments are on QD-LED EQE prediction. No evidence the approach generalizes to other properties, materials systems, or even other device metrics.

**Questions:**

Can you provide the ablation isolating whether gains come from adaptive sampling versus simply adding physics constraints?
For additional questions, please refer to the Weaknesses section.

---

### Official Review · Reviewer_Thre · 2025-10-30

**Soundness:** 2
**Presentation:** 2
**Contribution:** 2
**Rating:** 4
**Confidence:** 4

**Summary:**

This paper introduces reject sampling fine-tuning (RFT) to material discovery field, which aims to obtain high-quality CoT data. To meet the unique requirements in material discovery process, this paper proposes Physics-aware Gating and Adaptive Halting mechanism to filter low-quality and noisy data, thus finally resulting high-quality training data for material discovery SFT. Experimental results prove the effectiveness of the proposed data collection pipeline.

**Strengths:**

1. This paper successfully adopts RFT method to the material discovery domain.
2. The introduction of adaptive halting reduces the redundant computation as shown in Table 1.
3. The writing and structure of this paper are clear and easy to understand.

**Weaknesses:**

1. The novelty of this paper is limited. Essentially the work of this paper is simply adjusting the criteria of RFT from final answer correctness in math domain to EQE prediction in material discovery domain.
2. The evaluation of PaRS is also limited, which essentially only conduct ablation studies. It will be better to compare PaRS to existing SOTA methods of using LLMs for material discovery.

**Questions:**

See Weaknesses.

---

### Note · Authors · 2025-11-14

I have read and agree with the venue's withdrawal policy on behalf of myself and my co-authors.